# *Saccharomyces cerevisiae* as a Model System for Eukaryotic Cell Biology, from Cell Cycle Control to DNA Damage Response

**DOI:** 10.3390/ijms231911665

**Published:** 2022-10-01

**Authors:** Laura Vanderwaeren, Rüveyda Dok, Karin Voordeckers, Sandra Nuyts, Kevin J. Verstrepen

**Affiliations:** 1Laboratory of Experimental Radiotherapy, Department of Oncology, KU Leuven, 3000 Leuven, Belgium; 2Laboratory of Genetics and Genomics, Centre for Microbial and Plant Genetics, Department M2S, KU Leuven, 3001 Leuven, Belgium; 3Laboratory for Systems Biology, VIB-KU Leuven Center for Microbiology, 3001 Leuven, Belgium; 4Department of Radiation Oncology, Leuven Cancer Institute, University Hospitals Leuven, 3000 Leuven, Belgium

**Keywords:** DNA damage response, model system, *Saccharomyces cerevisiae*, yeast

## Abstract

The yeast *Saccharomyces cerevisiae* has been used for bread making and beer brewing for thousands of years. In addition, its ease of manipulation, well-annotated genome, expansive molecular toolbox, and its strong conservation of basic eukaryotic biology also make it a prime model for eukaryotic cell biology and genetics. In this review, we discuss the characteristics that made yeast such an extensively used model organism and specifically focus on the DNA damage response pathway as a prime example of how research in *S. cerevisiae* helped elucidate a highly conserved biological process. In addition, we also highlight differences in the DNA damage response of *S. cerevisiae* and humans and discuss the challenges of using *S. cerevisiae* as a model system.

## 1. Yeast: A Prime Eukaryotic Model System

Yeast has been indispensable to make beer and bread since 13,000–14,000 BC [1,2]. However, it was only in the 19th century that Louis Pasteur demonstrated the essential role of yeast in the fermentation process, where it is responsible for the conversion of cereal-derived sugars into ethanol and CO_2_ [3,4]. In 1883, Emil Christian Hansen was the first to isolate *Saccharomyces pastorianus*, the yeast responsible for fermentation of lager beers at the Carlsberg brewery [5]. Soon after, *Saccharomyces cerevisiae* was also isolated and quickly established itself as the most-commonly used yeast for the production of ale beers as well as a prime model for genetics and cell biology.

*S. cerevisiae* is a small (~5 µm) single-cell eukaryote and thus contains a nucleus and other membrane-bound organelles. Yeast cells are relatively easy to culture in laboratory conditions since they do not need a complex medium for growth. Moreover, cells divide rapidly, every 90 min, under optimal laboratory conditions through the process of budding, where a smaller, genetically identical daughter cell buds off the mother cell during mitosis. When, in 1978, Hinnen et al. published the successful transformation of yeast with a plasmid replicated in the bacterium *Escherichia coli*, yeast quickly became the most widely used single-cell eukaryotic model organism [6]. A plethora of replicating plasmids and selectable markers was developed. Moreover, apart from using plasmids, it quickly became clear that the efficient homologous recombination (HR) DNA repair machinery of *S. cerevisiae* allows integrating DNA fragments into specific genomic loci with an efficiency that is much higher compared to other organisms. Combined with the development of polymerase chain reaction (PCR), the efficient HR process enabled researchers to easily manipulate the yeast genome, which further established yeast as a model organism.

In 1996, the laboratory *S. cerevisiae* strain S288c became the first eukaryote for which the complete genome was sequenced. The results showed that a haploid S288c cell contains ~12,000 kilobases of genomic DNA, carrying approximately 6000 genes divided over 16 chromosomes [7]. This genetic architecture is specific for the S288c haploid lab strain that was sequenced. However, *S. cerevisiae* can exist in a haploid and diploid form. Moreover, naturally occurring *S. cerevisiae* strains are often genetically different and more complex, with a high degree of heterozygosity and frequently showing aneuploidy and polyploidy in their genomes [8,9,10].

Soon after publication of the yeast genome sequence, the yeast deletion collection was constructed. This collection consists of a nearly complete set of viable deletion mutants where, in each deletion mutant, a specific non-essential open reading frame is replaced with a drug resistance marker flanked by two distinct 20 basepair sequences or DNA barcodes, the UPTAG and the DOWNTAG, which are unique for each open reading frame [11,12] (Figure 1). Each UPTAG and DOWNTAG is flanked by universal primer sequences to allow PCR amplification of the barcode. The high efficiency of homologous recombination in yeast was exploited to integrate the marker cassette and made the construction of this deletion collection possible. The yeast deletion collection was constructed in the haploid S288c strain and consists of approximately 4800 deletion mutants. Essential genes cannot be deleted in this haploid strain as this would be lethal therefore a heterozygous diploid yeast deletion collection was also constructed [12]. Moreover, deletion collections were also constructed in other *S. cerevisiae* strains such as the Sigma1278b strain [13]. Two yeast strains can differ as much as two human individuals, as such genes essential in one genetic background can be dispensable in another [13].

The integration of DNA barcodes enables identification of individual mutants by their respective barcode, including tracking individual mutants in populations that contain a mix of all 4800 deletion mutants by deep-sequencing the barcode region from bulk samples, and then counting the relative proportion of each barcode [14,15,16,17,18]. The yeast deletion collection was crucial in research assigning specific cellular functions to genes. By growing the yeast deletion collection under different conditions, genes important for growth in particular conditions could be identified which significantly helped to assign the exact function of specific genes.

Several of the experimental procedures that were first developed in yeast cells, including barcoded siRNA screens [19,20] or CRISPR screens [21,22] were later also established in mammalian cells. However, these screens are often more complex compared to those in yeast. Other libraries such as the GFP library, in which each open reading frame is fluorescently tagged with GFP and a genome-wide over-expression library were constructed in *S. cerevisiae* to further characterize gene functions [23,24]. Furthermore, transcriptome analysis using DNA microarrays was also developed in yeast [25,26]. This was followed by more omics technologies such as proteomics and metabolomics as well as the development of methods such as ChIP-seq to map transcription factor binding [27,28,29,30,31].

Together, the molecular toolbox that was developed in *S. cerevisiae* and expanded to other organisms much accelerated research in cell biology, genetics, and genomics. Importantly, all information is collected and organized in the Saccharomyces Genome Database (SGD), a freely available online database which provides information on DNA and protein sequence, expression, regulation, interactions, phenotypes, Gene Ontology (GO) annotations, etc. for every yeast gene [32,33].

In 1987, it was shown that a human gene could complement a yeast mutant defective in the cell cycle [34]. This was a major breakthrough, since it illustrated that cell cycle control mechanisms were highly conserved between yeast and humans. In fact, despite being separated by approximately 1 billion years of evolution, more than one-third of the yeast genome has a homologous part in humans [35]. Furthermore, comparing the yeast genome sequence to that of other model organisms such as *Caenorhabditis elegans*, *Drosophila melanogaster* and the human genome sequence showed that protein amino acid sequences and protein functions have been conserved to such extent that GO annotations could frequently be transferred from one eukaryotic species to another. On average 32% amino acid identity is detected between yeast and human over the complete genome [36]. Additionally, approximately 50% of genes essential in yeast can be replaced by their human homolog [36].

The characteristics of yeast discussed above, namely the short life cycle, ease of manipulation, well-annotated genome, expansive molecular toolbox along with the strong conservation of basic eukaryotic biological and biochemical pathways, make yeast an excellent model organism to study eukaryotic cellular processes. Consequently, the yeast model has been used extensively in many fields, some of which will be shortly illustrated here.

There are many examples of key discoveries in the field of biology, genetics, biochemistry and medicine that have been made in yeast cells. A prime example is the discovery of key regulators of the cell cycle, for which the Nobel Prize was awarded in 2001 to Leland H. Hartwell, Tim Hunt and Sir Paul M. Nurse. Hartwell used *S. cerevisiae* to identify more than one hundred genes involved in cell cycle control, the cell division cycle or *CDC* genes [37,38,39]. He found that one of these genes, *CDC28*, controlled the first step in the progression through the G1 phase. He also introduced the concept of checkpoints as he discovered that the cell cycle is arrested upon DNA damage induced by X-ray radiation to allow time for repair [40]. This was later extended to multiple checkpoints to ensure a correct order of cell cycle phases [41,42,43,44]. Sir Paul M. Nurse used the yeast *Schizosaccharomyces pombe* as a model to identify the gene *CDC2* which was identical to *CDC28* identified by Hartwell [45,46,47,48,49]. The basic discoveries in yeast cells led to the hypothesis that defects in cell cycle checkpoints could be responsible for the uncontrolled growth and genomic instability of cancer cells [50]. An increased understanding of the cell cycle shed light on the molecular mechanisms for cellular transformation from normal to cancer cells which helped to identify targets for cancer therapies [51,52].

Other key eukaryotic cellular processes such as eukaryotic transcription, telomere structure, vesicle transport and autophagy were also unraveled using yeast as a model system, and several of the pioneering scientists were awarded Nobel prizes. In 2006, Roger Kornberg received the Nobel Prize in Chemistry “for his studies of the molecular basis of eukaryotic transcription” for deciphering the structure of the key components for transcription in yeast [53,54,55,56,57,58,59,60]. Jack Szostak, Elizabeth Blackburn, and Carol Greider used yeast to discover the telomerase function in eukaryotes and have received the Nobel Prize in Physiology and Medicine in 2009 [61,62,63,64,65,66,67,68,69,70]. In 2013 James E. Rothman, Randy W. Schekman and Thomas C. Südhof received the Nobel Prize “for their discoveries of machinery regulating vesicle traffic, a major transport system in our cells”. Schekman et al. used yeast to identify mutated genes that caused a defect in the transport machinery [71,72,73,74,75,76,77]. He then established an ordered secretory system and further elucidated the underlying mechanism. Later Rothman discovered a protein complex in mammalian cells that enables vesicles to dock and fuse with their target membranes [78,79]. Strikingly, some of the genes Schekman had discovered in yeast were homologous to those Rothman identified in mammals. This again demonstrated the conservation of a basic mechanism from yeast to man. In 2016 the Nobel Prize in Physiology or Medicine was awarded to Yoshinori Ohsumi “for his discoveries of mechanisms for autophagy” [80,81,82]. Ohsumi was the first to identify multiple yeast genes that regulate autophagy.

Yeast also serves as an important model for medical research. Almost 30% of known genes involved in human disease have yeast homologs [83,84]. Genome-wide approaches have been especially useful to screen for new biologically active compounds and to unravel drug-induced molecular mechanisms. The yeast deletion collection has been used to screen many different drugs to identify their molecular targets, including for example molecules that limit tumor growth such as Wortmannin and 5-Fluorouracil [85,86,87,88,89,90,91,92]. The heterozygous diploid yeast deletion collection has been useful in particular to screen for drug-induced haploinsufficiencies. This assay is based on the observation that a heterozygous deletion strain is sensitive to a drug that targets the protein expressed from the heterozygous locus [85]. The power of this assay lies in the simultaneous identification of both the inhibitory compound and its targets without prior knowledge of either of the two. The haploid yeast deletion collection cannot be used to identify targets of a certain drug, because the target is absent. However, the deletion collection is particularly useful for identifying genes that act to buffer the compound target pathway and are thus required for growth in presence of the compound. As a result, this assay can be used to screen compounds that do not directly target a protein, such as DNA damaging agents or reactive oxygen species (ROS)-inducing compounds, and identify genes important for the response to these agents [14,93].

Central to all these discoveries is the high conservation of basic biological processes between yeast and humans. One of such highly conserved pathways is the DNA damage response, which makes it possible to use yeast as model system to increase our basic understanding of the DNA damage response. In the following sections, the highly conserved DNA damage response of *S. cerevisiae* is described and compared to humans.

## 2. The DNA Damage Response in *S. cerevisiae* and Humans

DNA is susceptible to damage by endogenous as well as external factors such as ultraviolet (UV) light, ionizing radiation, and alkylating agents. Damage induced in the DNA can be in the form of single-strand breaks (SSB) and double-strand breaks (DSB). DSBs are considered most harmful since this type of lesion is less efficiently repaired. Upon DNA damage, the DNA damage response is activated to repair the DNA and promote cell survival. In the next sections, the DNA damage response and the major SSB and DSB DNA repair pathways of *S. cerevisiae* are described.

The core DNA repair process of the different pathways are largely similar, with 70% of yeast DNA repair proteins having a human homolog. However, DNA repair mechanisms in humans often involve a larger number of proteins and components. Here, we do not describe human DNA repair pathways in full detail, but homologs of the yeast proteins are shown in the figures and relevant differences in DNA repair between yeast and humans are highlighted. Examples illustrate how *S. cerevisiae* has been used as a model organism.

### 2.1. Signaling of DNA Double-Strand Breaks

In *S. cerevisiae*, signaling of DNA damage depends on the kinases Mec1 and Tel1 (Figure 2). DSBs are recognized by the MRX complex (composed of Mre11, Rad50 and Xrs2), which recruits and activates Tel1 [94,95]. Tel1 in turn activates the effector kinase Chk1. Mec1 is recruited to stretches of ssDNA bound by replication protein A (RPA, a heterotrimeric protein formed by the subunits Rfa1, Rfa2 and Rfa3) through its interaction partner Ddc2, where it phosphorylates and activates Rad53 together with Dpb11 which is loaded onto ds/ssDNA junctions by the 9-1-1 complex, composed of the proteins Ddc1, Rad17, and Mec3. Both Tel1 and Mec1 can phosphorylate Chk1, Rad9, and Rad53 directly and can as well phosphorylate lesion-proximal substrates such as histone H2A. Rad9 can bind these histone modifications and serve as an adaptor protein for Rad53 activation [96,97]. The activated DNA damage signaling kinases Chk1 and Rad53 then mediate the responses to DNA damage that include cell cycle arrest, activation of DNA repair pathways, inhibition of origin firing, protection and restart of stalled replication forks, initiation of apoptosis and control of dNTP levels [98,99,100]. Rad53 can further phosphorylate and activate Dun1, which upregulates the transcription of a specific set of DNA damage-induced genes, including subunits of ribonucleotide reductase (RNR) [99,101,102]. Human homologs of the sensing kinases Mec1 and Tel1 are ATR and ATM, respectively. Their function and phosphorylation cascade are largely similar to the process in yeast; however, a DUN1 homolog has not been identified. Instead activation of ribonucleotide reductases is performed by p53, which is a target of the checkpoint kinases CHK1 and CHK2, the mammalian homologs of Chk1 and Rad53 [103].

### 2.2. Single-Strand Break Repair Mechanisms

#### 2.2.1. Base Excision Repair

DNA lesions that are not associated with structural alterations of the DNA helix such as oxidized, deaminated and alkylated bases or apurinic/apyrimidinic sites (AP sites) are repaired by base excision repair (BER) [104]. In *S. cerevisiae*, BER occurs in two stages (Figure 3). First, a damage-specific step depends on DNA glycosylases that recognize a specific type of base lesion. *S. cerevisiae* contains six DNA glycosylases: monofunctional Ung1, Mag1, and Mag2 and bifunctional Ntg1, Ntg2, and Ogg1, each targeting specific damage [105,106,107,108,109]. For example, Mag1 recognizes alkylated adenine bases, whereas Ogg1 targets 8-oxoguanine. The glycosylases bind the minor groove of the DNA, thereby inducing a kink. Next they flip the abnormal base out of the DNA helix and cleave the β glycosidic bonds between the ribose and base [110]. The resulting 3′ deoxyribose phosphate is then processed by AP endonucleases [111]. In *S. cerevisiae* two AP endonucleases are known; Apn1 and Apn2 [112,113]. The 3′ OH site serves as the initiation site for DNA synthesis and the 5′ diphosphate (5′-PP) end facilitates DNA polymerase binding [114].

There are two subpathways of BER: short and long patch repair. In short patch repair, one single nucleotide is excised. Gap filling and ligation is performed by polymerase β and the ligase Cdc9. In long patch repair, polymerase δ and Pol30 are responsible for gap filling with strand displacement resulting in 2–13 nucleotide being replaced [115]. This process creates a 5′ overhang tail, which is cleaved by the flap endonuclease Rad27 [116]. The BER pathway is largely similar in humans; homologs of the yeast proteins are listed in Figure 3. In addition, an important additional protein, poly(ADP-ribose)polymerases 1 (PARP-1), functions in human BER but does not have a yeast homolog [117]. PARP-1 recognizes and binds to DNA SSBs and recruits other BER repair proteins such as DNA [118,119,120] polymerase β and the XRCC1-DNA ligase III complex to the site of damage [121].

As BER was first described in *E. coli*, yeast has proven particularly useful in the translation of the BER pathway from a prokaryotic to a eukaryotic system. This is illustrated here with example of the identification of *OGG1*. In *E. coli*, the genes *Mutm* and *Fpg* are two DNA glycosylases that prevent spontaneous mutagenesis [122]. Consequently, inactivation of both *mutM* and *Fpg* results in a mutator strain which was used to identify the *OGG1* gene from *S. cerevisiae* in a functional complementation assay. Therefore, the *E. coli* strain lacking *mutM* and *Fpg* was transformed with a yeast DNA library and clones that showed a reduced spontaneous mutagenesis were selected. Subsequent sequencing and characterization identified the *OGG1* gene on chromosome XIII [105]. Scanning the human sequence databases for homology to the *S. cerevisiae OGG1*, the human *OGG1* gene was rapidly identified. A similar approach using a human genomic library would not have been possible, as the Ogg1 protein had no sequence similarity to the *E. coli Fpg* or *MutM*. As such cloning strategies based on sequence similarity would not have been successful and a functional complementation assay was necessary to identify *OGG1*. However, in *E. coli* functional complementation is less successful for human genes as *E. coli* lacks the mRNA splicing machinery. *S. cerevisiae* on the other hand harbors relatively few introns. As such a yeast genomic library can be used for expression in *E. coli.* This example illustrates the power of yeast as tool to identify the BER genes.

#### 2.2.2. Mismatch Repair

Ensuring correct DNA replication is essential for maintaining genomic stability. It has been estimated that a wrong nucleotide is incorporated once every 10^4^ to 10^5^ nucleotides [123]. However, the observed final error rate after DNA replication is much lower, 0.33 × 10^−9^ per site per cell division for *S. cerevisiae* and 0.1 × 10^−9^ per site per cell division for humans [124]. This suggests the existence of replication and post-replication DNA repair mechanisms that correct part of the erroneous incorporations. Firstly, high-fidelity DNA polymerases possess proofreading activity that can remove wrongly incorporated bases. Second, DNA mismatch repair (MMR) is a highly conserved DNA repair pathway that corrects mismatches that have escaped the proofreading activity of the polymerases. The MMR repair pathway was extensively studied in *Escherichia coli* and homologs were later identified and further characterized in eukaryotes, including yeast [125,126].

In *S. cerevisiae,* the different steps in the mismatch repair pathway include mismatch recognition by the yeast MutS and MutL homologs [127,128,129], excision of a region that contains the mismatched base by the exonuclease Exo1 [130,131,132] and DNA re-synthesis followed by ligation (Figure 4). In eukaryotes MutS is a heterodimeric protein that can consist of either Msh2/Msh6 (MutSα) or Msh2/Msh3 (MutSβ). Substrate recognition by the MutS homologs is followed by a conformational change of MutS homologs driven by ATP hydrolysis [133,134]. MutS changes into a sliding clamp and recruits the heterodimer of MutL homologous [135]. Here too there are several MutL homologs: MutLα, consisting of Mlh1/Pms2 and MutLβ formed by Mlh1/Mlh3. The complex that is now formed by MutS and MutL homologs can slide over the DNA in both directions due to the sliding clamp activity of MutS. MutL is essential for the recruitment of downstream MMR factors to the lesion [136].

The sliding continues until the complex encounters a strand discontinuity or nick (for example a gap between Okazaki fragments) that is bound by the DNA polymerase processivity factor proliferating cell nuclear antigen (PCNA) encoded by *POL30* in yeast [137,138]. PCNA is important to link the MMR complex to DNA polymerase at the replication fork, ultimately facilitating the distinction between daughter and template strand [139]. When MutS-MutL encounters PCNA, this initiates loading of the exonuclease Exo1 [130,131,132]. Exo1 then starts degrading the nicked strand and thereby removes the mismatch. The single stranded gap in the DNA is stabilized by RPA and filled in by polymerase δ using the parental DNA strand as a template. Lastly the DNA is ligated by Cdc9 in [140,141].

#### 2.2.3. Nucleotide Excision Repair

Nucleotide excision repair (NER) is a DNA repair mechanism that recognizes and repairs bulky DNA damage (Figure 5). Depending on where in the genome the lesion is located, two different subpathways of NER recognize the lesion, namely global genome nucleotide excision repair (GG-NER) and transcription-coupled nucleotide excision repair (TC-NER). In GG-NER damages in the entire genome, including non-transcribed regions and silent chromatin, are repaired. TC-NER on the other hand is responsible for the repair of lesions in the transcribed strand of active genes.

In GG-NER, the Rad23-Rad4 complex continuously scans the DNA until a lesion is recognized [142]. When the Rad23-Rad4 complex actually recognizes a damaged DNA site, a more stable Rad23-Rad4-DNA complex is formed and downstream NER factors are recruited to the site of damage.

As the name suggests, transcription coupled (TC) NER is initiated when damage to a transcribed DNA strand limits transcription activity of that strand. The stalled RNA polymerase serves as DNA damage recognition signal during transcription. The lesion-stalled RNA polymerase is recognized by the TC-NER-specific protein Rad26, which then recruits Rad28 that is required for further assembly of the NER complex [143,144,145].

After damage recognition, the transcription initiation factor IIH (TFIIH) complex together with the proteins RPA and Rad14 are recruited to the DNA damage site in both TC-NER and GG-NER. TFIIH is a multisubunit complex composed of two helicases (Ssl2 and Rad3), the complex of CDK-activating kinase (CAK: Kin28 and Ccl1) and structural proteins (Tfb1, Tfb2, Tfb3, Tfb4, Tfb5 and Tfb6) that form the core. The structural proteins have no enzymatic activity. The two helicases have opposite polarities (Ssl2 3′–5′, Rad3 5′–3′), which results in an extension of the open DNA configuration around the lesion, forming a DNA bubble. Rad14 stimulates TFIIH helicases activity, while RPA stabilizes the ssDNA by binding to it. On top of its helicase activity, Ssl2 also possesses ATPase activity, which enables the recruitment of the TFIIH factor [145,146,147].

Next, dual incision is performed by two structure-specific endonucleases Rad2 and Rad10-Rad1. The endonucleases respectively cut the damaged strand 3′ and 5′ from the lesion, leaving a single-strand gap of 22–30 nucleotides [145,148,149].

RPA and Rad2 further coordinate the synchronization of lesion excision and DNA gap filling, which prevents the accumulation of ssDNA gaps that may induce DNA damage signaling. Directly after the 5′ incision Pol30 (PCNA) is loaded and recruits a DNA polymerase (DNA Pol δ, DNA Pol κ or DNA Pol ε) to fill the gap in the DNA [150]. Finally, DNA ligase Cdc9 seals the nick [151].

In humans GG-NER is performed by the XPC complex (composed of XPC, Rad23b, and Cetn2). However, the XPC complex is inefficient in recognizing the typical lesions induced by UV light exposure, namely cyclobutane pyrimidine dimers, due to their low degree of structural perturbation [152]. Consequently, these lesions are recognized by an alternative complex, namely the UV-DDB complex. This complex is composed of two DNA damage binding proteins, DDB1 and DDB2. These two proteins associate with the CRL complex composed of CUL4A and RBX1 (ROC1). The newly formed complex aids in remodeling the damaged site to prepare it for subsequent NER reactions by ubiquitination of histones H3, H4 and H2A [153,154,155]. These ubiquitination events enhance the recruitment of XPC to the site of damage and thereby facilitate the repair process. DDB2 binds to the UV-induced damaged DNA, extrudes the lesion into its binding pocket and kinks the DNA [156]. This action of DDB2 creates ssDNA, which facilitates XPC binding.

NER has been extensively studied in mammals as deficiency in NER is associated with multiple syndromes such as xeroderma pigmentosum and Cockayne syndrome. As such studies in cell lines of xeroderma pigmentosum patients have enhanced our knowledge about NER [157]. Nonetheless, genetic and biochemical studies in yeast made major contributions in elucidating the molecular mechanisms of NER. For example, studies with a *rad1* mutant allele, which encoded a protein that could not interact with Rad10, revealed that complex formation was essential for the functioning of these proteins [158].

### 2.3. Double-Strand Break Repair Mechanisms

#### 2.3.1. Homologous Recombination

HR is a DSB repair pathway conserved from bacteria to humans. In yeast it the most-used DSB repair pathway (Figure 6). This DNA repair pathway allows accurate, high-fidelity repair of DSBs because it uses the undamaged sister chromatid as a repair template. As yeast transformations rely on HR, this pathway was extensively studied in this model organism, thereby using the unique properties of the *MAT* locus and *HO* endonuclease.

Depending on the allele present in the *MAT* locus, haploid *S. cerevisiae* can exist in two mating types, **a** or α (diploid cells typically carry one **a** and α allele). Via homologous recombination initiated by a DSB at the *MAT* locus, haploid yeast cells can autonomously change mating type. The introduction of this DSB is catalyzed by the endogenously encoded *HO* endonuclease. Mating type switching then occurs via homologous recombination using the opposite mating type information, present at the *HML* and *HMR* loci, as a template [159]. In normal conditions, the *HO* gene is tightly regulated. However for DSB repair studies, a galactoses-inducible *HO* gene was engineered [160]. This enabled to express *HO* in all cells at the same time and thus to introduce a DSB at the *MAT* locus to all cells simultaneously by growing the yeast cells on galactose [161,162]. Further engineering the recognition sequences allowed to introduce a DSB at different loci in the genome, which facilitated the unraveling of the mechanistic details of the process [163,164,165]. Furthermore, by analyzing the repair kinetics and intermediates in strains where genes involved in homologous recombination were mutated or deleted, the role of these genes in homologous recombination could be investigated [166,167,168]. Moreover, immunostaining and chromatin immunoprecipitation (ChIP) protein recruitment to a double-strand break could be assayed [169,170,171,172].

The model for repairing broken ends using a homologous template was first proposed by Resnick in 1976 [173]. In a first step, the DSB is pre-processed to a 3′ overhanging tail by extensive DNA end resection [174]. First, initial end processing is performed by the MRX complex (Mre11, Rad50 and Xrs2) together with Sae2, followed by more extensive resection by the exonuclease Exo1 or the helicase Sgs1 together with the nuclease Dna2 [175,176]. The resulting single stranded DNA (ssDNA) is bound by RPA, a ssDNA stabilizing protein. Next, Rad51 needs to replace RPA in order to form the Rad51 filament. As RPA is an abundant protein and binds ssDNA with very high affinity, recombination mediator proteins, Rad52 and Rad55-Rad57, are needed to help Rad51 replace RPA [172,177,178,179,180]. When Rad51 is completely loaded onto the ssDNA, the presynaptic filament is formed.

The presynaptic filament, facilitated by Rad54, scans the genome for sequence complementarity in order to find a homologous part from which to initiate DNA synthesis [181,182]. When a homologous donor dsDNA is found, the synaptic complex is formed. The intermediate that is formed by strand invasion is called a displacement loop or D-loop, because the original base pairs in the donor dsDNA must be disrupted and replaced by base pairing of the invading strand with the donor. The D-loop can migrate in the DNA via replication by DNA polymerase δ [183]. Before DNA synthesis can start, Rad51 must be dissociated from the heteroduplex DNA. This is mediated by Rad54 [172,181,182]. Next, the second end of the DSB is engaged by Rad52 in a process called second end capture, which results in the formation of a double Holliday junction [184,185,186]. The double Holliday junction is resolved by the combined action of the helicase Sgs1, the topoisomerase Top3 or the Mus81/Mms4 complex resulting in crossover and non-crossover products [187,188,189]. In crossover products the part of the homologous strand used as a template is exchanged. In non-crossover products this is not the case.

In humans, the MRN complex together with CtIP initiate limited end resection which is further extended by BLM helicase together with EXO1 and DNA2 exonucleases. RPA than binds to the single stranded DNA ends and is replaced by Rad51, which in humans in mediated by BRCA2.

#### 2.3.2. Non-Homologous End-Joining

Non-homologous end-joining (NHEJ) is a second type of DSB repair pathway that directly ligates the DNA ends together in an error-prone manner (Figure 7). Contrary to yeast, where HR is the most used DSB repair pathway, in humans NHEJ is the most common pathway to repair DSBs.

NHEJ is initiated by binding of the Ku protein, a heterodimer composed of the subunits Yku70 and Yku80, and the MRX complex to the DNA ends of the DSB [190]. This binding blocks resection and prevents formation of long ssDNA tracts necessary for initiation of HR [191,192]. The role of the MRX complex, which also functions in HR, is not well understood. It is thought that MRX functions as a complex that bridges DSB ends and acts as a flexible tether to assist ligation in NHEJ [193]. Tethering and protection of the ends stabilizes the ends and brings them together. The Ku complex then recruits other core NHEJ proteins, such as Dnl4-Lif1 and Nej1, to the DSB site forming the repair super complex. End-processing factors such as Pol4 act in parallel with the complex formation and are required to make complementary ends needed for ligation. Pol4 is known to initiate gap filling at unstable 3′ overhangs [194,195]. After end-processing ligation is carried out by DNA ligase IV, Dnl4 in *S. cerevisiae*, stabilized by Lif1 [196,197]. To ensure stable formation of the NHEJ super complex at DSB ends, Nej1 functions as a stabilizing factor [198].

Even though the process of NHEJ is largely similar between yeast and humans, the initiation of NHEJ is performed by different proteins. In humans, Ku70/Ku80 heterodimers recruit DNA-dependent protein kinase catalytic subunit (DNA-Pkc) which undergoes auto-phosphorylation and thereby recruits other repair factors [199]. Next, end processing is performed by the Artemis endonuclease [200]. For both DNA-Pkc and Artemis, no homologous have been identified in *S. cerevisiae*.

Repair pathways for different types of base damages, such as BER, NER, and MMR rely on specialized enzymes for the recognition of the type of damage. However, the repair of DSBs can be performed by both HR and NHEJ, and the choice of which DSB repair pathway is activated depends on regulatory processes that are influenced by the cell type, the structure of the DNA ends resulting from resection, as well as the phase of the cell cycle. In both yeast and humans, HR is most active in the S and G2 phase when homologous templates are available. NHEJ on the other hand is active during the complete cell cycle [191,201,202]. This indicates that HR and NHEJ compete at DSBs. The choice between HR and NHEJ is then controlled by the extent of 5′–3′ end resection determined by the Ku heterodimer, the MRX complex, and the endonuclease Sae2. HR starts with extensive end resection of the 5′ strand to generate ssDNA which is bound by RPA. Once resection has started, the long ssDNA ends become poor substrates for Yku70 and Yku80 [203].

Since yeast is less efficient in NHEJ, NHEJ was only observed in strains lacking genes involved in HR [204]. Moreover, deletion of yeast NHEJ genes in the presence of functional HR genes does not result in sensitivity to several DNA damaging agents. This makes studying NHEJ in yeast more difficult using conventional molecular and genetic assays. Nonetheless, elegantly designed assays can overcome this problem. Ooi et al. for example, used an in vivo plasmid repair assay to screen the yeast deletion collection for genes involved in NHEJ. In their assay, they transformed the yeast deletion collection with circular and linearized plasmids and measured the transformation efficiency as a quantitative readout. Deletion mutants with an active NHEJ process will be able to repair the linearized plasmid and grow to similar levels as when they are transformed with a circular plasmid. Deletion mutants with a defect in NHEJ fail to recircularize the plasmid which will result in a lower transformation efficiency. Using this assay, they were able to identify the gene *NEJ1* [205].

## 3. Challenges and Future Perspectives of Yeast as a Model System

Multiple DDR kinases, such as Dun1, Mec1, Chk1, and Rad53, were first identified and characterized in yeast [44,101,206,207,208]. Additionally, genome-wide screens of the yeast deletion collection with DNA damage inducers such as MMS, UV light or ionizing radiation further continued to identify genes involved in the DNA damage response [93,209,210]. Many of the genes identified in yeast proved to be conserved in higher eukaryotes, and as such, the yeast work much facilitated the unraveling of DNA damage response mechanisms in other organisms. However, using yeast as a model organism for higher, more complex organisms does come with some pitfalls and thus results cannot always be translated without verification. Most studies in yeast are performed in haploids even though more complex organisms such as humans are diploid. Haploid genomes are easier to manipulate and make it easier to see the effect of certain mutations, deletions, etc. However, diploids and haploids can show a different regulation of certain DNA repair pathways. For example, diploids yeast cells are known to exhibit lower levels of NHEJ than haploid cells [211,212,213,214].

Moreover, pathways for double-strand break repair are differentially favored in yeast compared to humans. Early experiments in yeast, where cell killing by radiation was investigated, seemed to indicate that HR was the most dominant DSB repair mechanism in yeast. The type of DNA ends produced by radiation do not form a suitable template for NHEJ as they first require processing by the MRX complex. In mammals on the other hand, NHEJ is the preferred pathway for repair of DSBs. It has been observed that the process of NHEJ is faster in than HR in humans [201]. Moreover, due to the highly repetitive nature of human genomes, efficient HR could lead to deleterious genomic rearrangements.

Why yeast NHEJ is so inefficient is unclear. The most likely reason is that mammals have more NHEJ proteins for which no homolog is present in yeast [215]. Although *S. cerevisiae* possesses all core NHEJ factors, it lacks the DNA-dependent protein kinase catalytic subunit (DNA-PKcs) that in mammals is recruited to DSBs by the Ku complex and is required for NHEJ. As a result, the mammalian DNA damage response is coordinated by three DNA damage sensors: ATM, ATR, and DNA-PK. In yeast, the MRX complex compensates for the lack of DNA-PK, but perhaps this does result in lower NHEJ activity [190,193]. This difference between NHEJ in yeast and human illustrates the more complex nature of the mammalian DNA damage response. Although the core processes of the DNA damage response are largely similar, in mammals, much more genes are involved in the DNA damage response. To elucidate the functions of all these genes, more research is required.

Since the toolbox for studying higher eukaryotes steadily expands, including, for example, the implementation of various technologies based on CRISPR-Cas, the need to use a model system may appear to decline. However, the yeast continues to be an important model system to further unravel the full functioning and coordination of the DNA damage response. It has become clear that DNA damage response kinases also regulate other processes such as carbon metabolism, autophagy, and protein homeostasis [216,217,218,219]. Moreover, the list of genes that are shown to play a role in the DNA damage response is still steadily increasing [220]. For example, the recent focus on phosphoproteomics signaling identified new targets of the DNA damage kinases [221,222]. Moreover, the exact molecular function of specific DNA damage response proteins is also being investigated in yeast [223,224].

These new studies in the field of DNA damage response also result in the development of multiple new techniques. For example, in order to study the human DNA repair proteins using a yeast system, humanized yeasts form an excellent tool. Many protein-coding human genes can successfully substitute for their yeast equivalents and sustain yeast growth [36,225,226]. This allows us to study the function of human proteins in a less complex organism and to characterize several cancer-associated mutations and their influence on protein functioning [227,228,229].

In addition, Duan et al. studied the genome-wide role of Rad26, the protein that initiates TC-NER, using chromatin immunoprecipitation sequencing (ChIP-seq) [230]. Peritore et al. used strand-specific ChIP-seq to determine which DNA repair proteins associated with the ssDNA and dsDNA, respectively, at a DSB site. They were able to confirm the ssDNA-binding nature of RPA and Rad51 but also visualized the dsDNA-binding nature of Rad51 during the homology search. Moreover, they showed that nucleosomes are removed during the resection process and are thus not associated with ssDNA at the site of damage [231].

Alternative versions of the inducible HO-system have also been developed. One of the shortcomings of the galactose inducible HO-system is that it requires specific nutrient conditions for its induction. This limits the metabolic states in which DSB signaling and repair can be studied. Moreover, efficient galactose-induction requires cells to be precultured in a non-fermentable carbon source. This can become tedious in the case of DSB repair mutant strains since they often have a defect in cell growth. To overcome these limitations, recently a media-independent heterologous induction system that controls HO expression was developed. Using β-estradiol they were able to induce a single DSB in an efficient manner under different carbon sources such as glucose, lactic acid and galactose [232]. Additionally, using several restriction enzymes that specifically recognize between 20 and 100 8 bp recognition sites in the yeast genome, they were able to develop an inducible system that results in multiple DSBs at defined loci [232].

Technologies based on CRISPR-Cas have emerged as a powerful genome editing tool. The outcome of the CRISPR-Cas-induced double-strand break can vary depending on the repair pathway that is used. While NHEJ results in insertions or deletions at the targeted locus, HR results in precise mutations or complete repair by using the homologous sequence that is provided. Vyas et al. showed that different yeasts rely differentially on either HR or NHEJ for the repair of the CRISPR-Cas9-induced DSB. While *S. cerevisiae* and *Candida albicans* rely on HR, *Candida glabrata* and *Naumovozyma castellii* mainly rely on NHEJ [233]. Consequently, because different organisms have different primary repair pathways, specific genetic editing needs need to be taken into account in order to optimize CRISPR-Cas9 mutagenesis. Lemos et al. studied how binding of the Cas9::gRNA complex influenced DSB repair by NHEJ. Therefore, they assessed the repair profiles of gRNAs pairs that were complementary to opposite DNA strands and cleaved at the same chromosomal location. They repeatedly found insertions of 1 base, in which the added base was dependent on the orientation of Cas9::gRNA binding [234]. Studying the repair of CRISPR-Cas induced DSBs in yeast can further improve the development of this genome editing tool.

In addition, in radiation biology, yeast served as an important model organism to study the response to radiation. As previously mentioned, Hartwell introduced his concept of checkpoints based on observations made after radiation treatment [40]. In addition, multiple DNA damage response genes have been identified using radiation to induce DNA damage; hence, many of these genes are termed “rad”-genes [206,235,236]. Even now, with the increased toolbox for higher eukaryotes, yeast remains a useful model in radiation biology. For example, to investigate radiation damage and repair after different radiation modalities, such as conventional photon radiotherapy and proton radiotherapy [15,237].

Together, it is clear that despite the shortcomings and pitfalls, yeast remains an important model system to further unravel the molecular mechanisms of proteins functioning in the DNA damage response, as well as to further explore the coordination and regulation of the DNA damage response. As such, the simplicity of the yeast model remains key in unraveling and understanding a complex pathway as the DNA damage response.

## 4. Conclusions

Besides its indispensable role in food production, the yeast *S. cerevisiae* proved an excellent model organism to study basic eukaryotic biology, as exemplified by the long list of Nobel prizes that have been awarded to yeast researchers. The existing molecular toolbox, together with the extensive genetic and physiological knowledge and the high conservation of core processes, make *S. cerevisiae* a particularly suitable and popular model for to study of eukaryotic biology. New techniques and approaches for studying biological processes, such as different omics approaches, keep being developed and continue to increase our knowledge.

The DNA damage response is a prime example of a eukaryotic pathway in which the key steps were first unraveled using yeast as a model organism. Especially HR was extensively studied in yeast using the HO locus to introduce one single defined double-strand break. In addition, genome-wide screens using the yeast deletion collection identified multiple genes involved in the DNA damage response to different DNA damage inducers.

It should be noted, however, that using yeast as a model for more complex organisms comes with pitfalls. However, the yeast model remains key in unraveling and understanding complex pathways, as it seems that new technologies and findings often still first emerge in this fantastic organism.

## Figures and Tables

**Figure 1 ijms-23-11665-f001:**
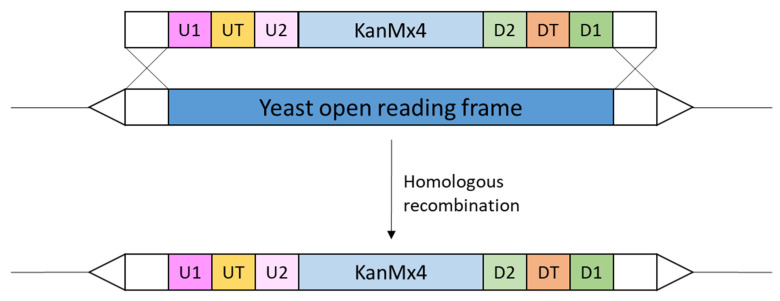
Schematic overview of the construction of the yeast deletion collection. The cassette used to replace and thus delete every yeast open reading frame consists of a kanamycin-resistance gene (KanMx4) flanked by two 20 basepair sequences, called DNA barcodes, the UPTAG (UT) and the DOWNTAG (DT), which are unique for each gene. Each UPTAG and DOWNTAG is flanked by universal primer sequences (U1 and U2 for UPTAG, D1 and D2 for DOWNTAG) to allow PCR amplification of the barcodes. The DNA 5′ and 3′ to the barcodes is homologous to the yeast DNA flanking the yeast open reading frame (indicate by crosses). Yeast incorporates the cassette through homologous recombination which results in the replacement and thus deletion of the open reading frame by the cassette sequence.

**Figure 2 ijms-23-11665-f002:**
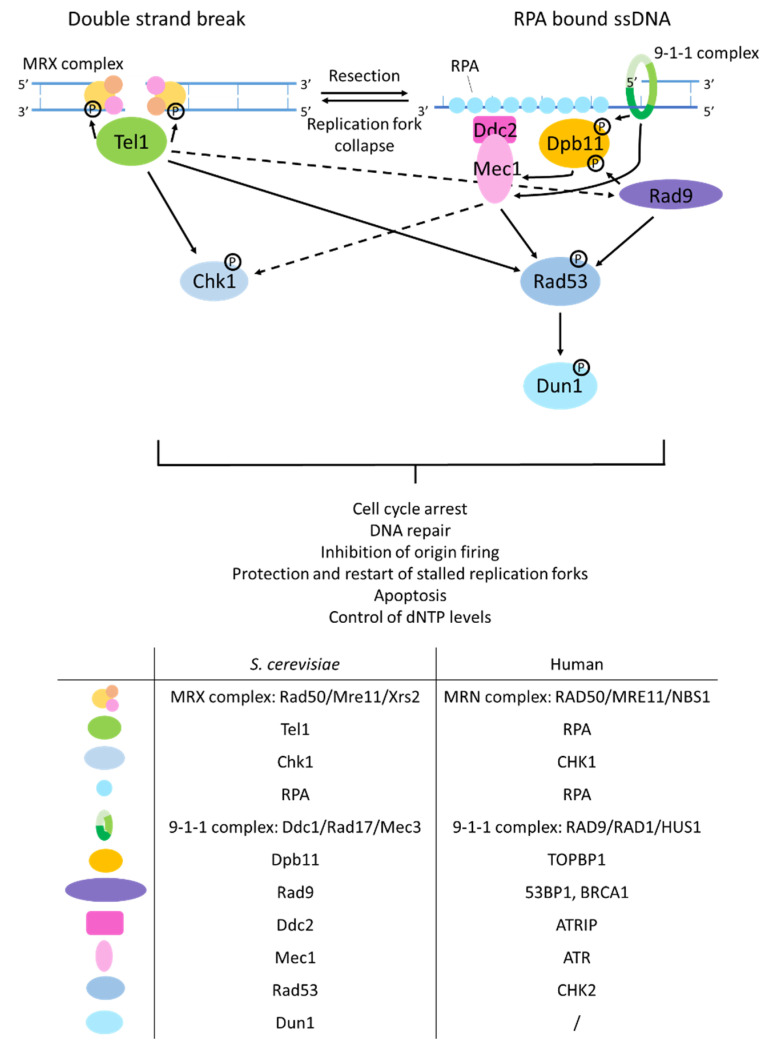
DNA damage signaling pathways. Upon DNA damage two sensor kinases Tel1 and Mec1 start a phosphorylation cascade. Double-strand breaks (DSB) are recognized by the MRX complex, which in turn is recognized by Tel1. Tel1 then phosphorylates Chk1. ssDNA stretches bound by RPA are recognized by Ddc2. Together the 9-1-1 complex and Dpb11 activate Ddc2-Mec1. Activated Mec1 phosphorylates Rad53, which in turn phosphorylates Dun1. Rad9 serves as an adaptor protein to activate Rad53. Crosstalk between Tel1 and Mec1 can occur when a stalled replication fork collapses which can result in a DSB or when a DSB is resected which results in ssDNA stretches. Human homologs of the different yeast proteins are listed in the table.

**Figure 3 ijms-23-11665-f003:**
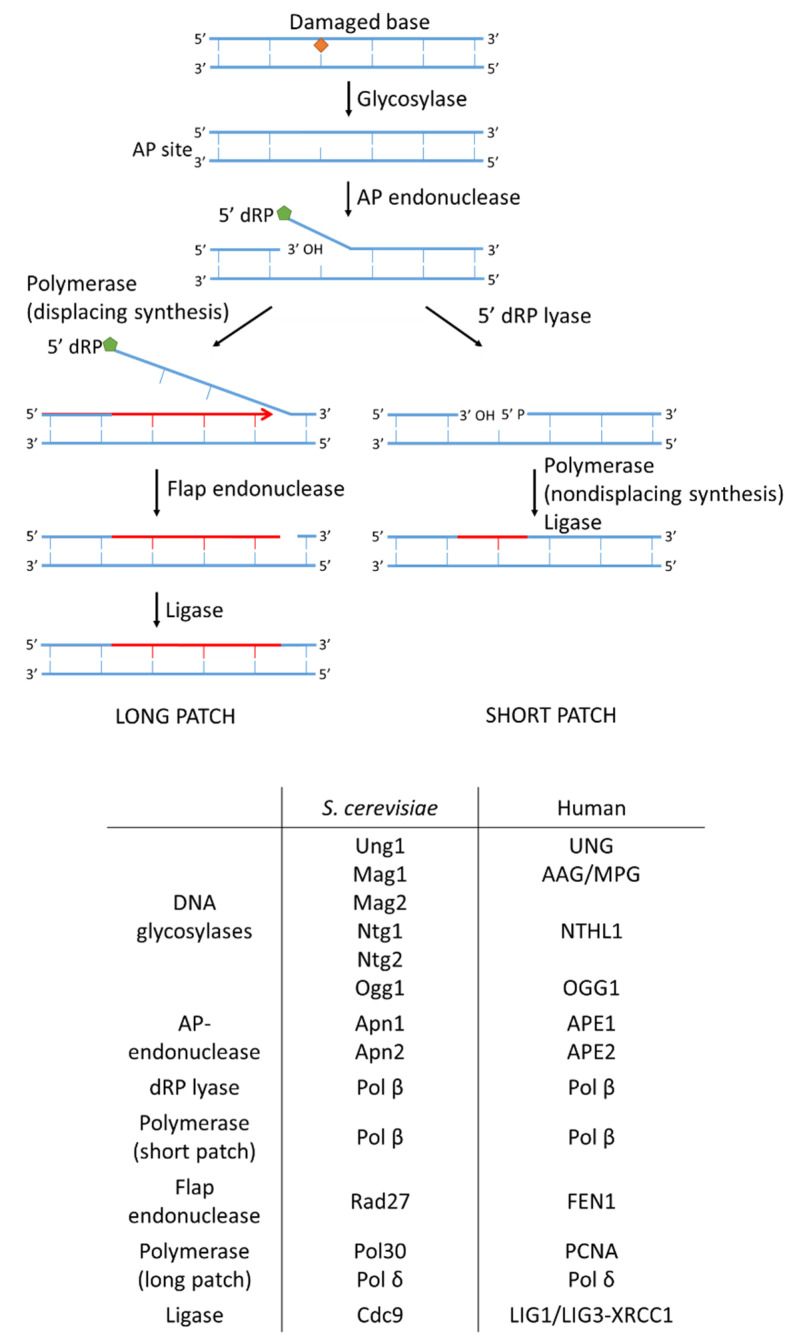
Mechanisms of base excision repair (BER), mismatch repair (MMR), and nucleotide excision repair (NER). (A) Mechanism of BER. The base lesion is recognized and removed by DNA glycosylases. The resulting apuric or apyrimidic (AP) site is processed by AP endonucleases. In a long patch, BER displacing synthesis is followed by flap removal by Rad27, and ligation takes place. In a short patch, BER deoxyribose phosphate (dRP) lyase excises one nucleotide, which is followed by gap filling and ligation.

**Figure 4 ijms-23-11665-f004:**
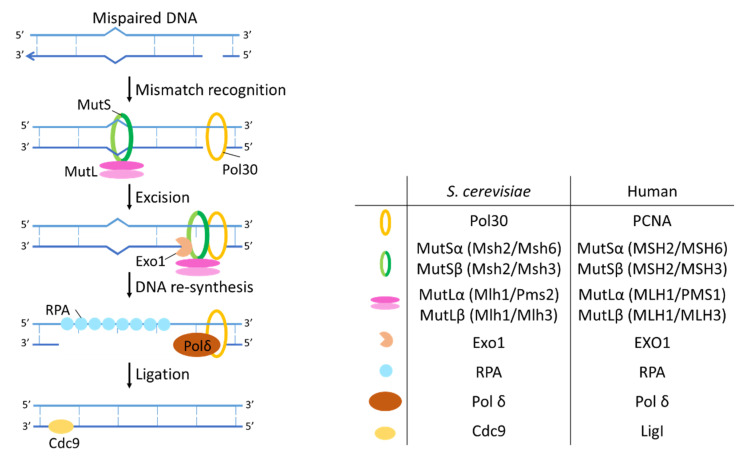
Mechanism of MMR. MutS and MutL homologs recognize the mismatch followed by excision of the mismatch by Exo1 and DNA re-synthesis and ligation.

**Figure 5 ijms-23-11665-f005:**
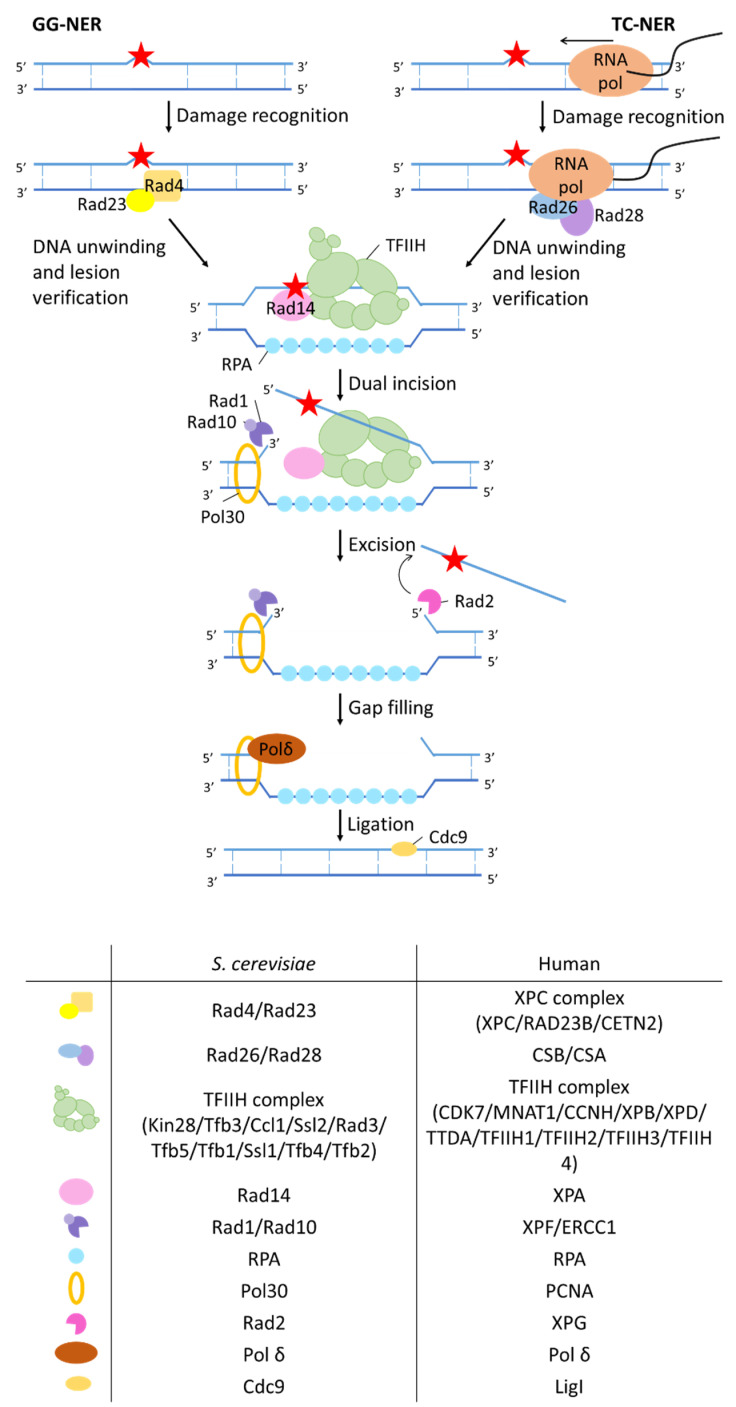
Mechanism of NER. In global genome (GG) NER, the Rad4/Rad27 complex recognizes the lesion. In transcription-coupled (TC) NER, the Rad26/Rad28 complex recognizes the stalled RNA polymerase. Next the transcription initiation factor IIH (TFIIH) complex is recruited. The helicase Ssl2 and Rad3 within the TFIIH complex extend the open DNA configuration around the lesion with their opposite helicase polarity. Rad14 stimulates TFIIH helicases activity while RPA stabilizes the ssDNA by binding to it. Dual incision is performed by two structure-specific endonucleases Rad2 and Rad10-Rad1 which is followed by gap filling and ligation. Human homologs of the different yeast proteins are listed in the tables.

**Figure 6 ijms-23-11665-f006:**
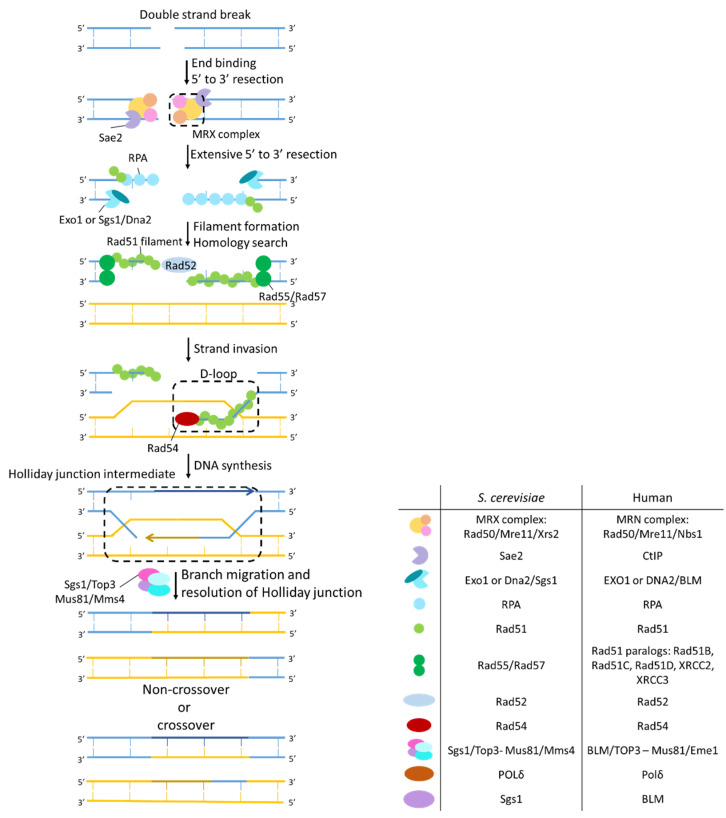
Mechanism of homologous recombination (HR). DSBs are recognized by the MRX complex. Sae2 catalyzes initial end processing, after which Exo1, Sgs1, and Dna2 perform extensive end resection to form long stretches of ssDNA bound by RPA. Rad51 mediated by Rad52 and Rad55-Rad57 replaces RPA to form the Rad51 filament. Rad54 helps the Rad51 filament to find homologous sequences. When a homologous sequence is found, strand invasion is performed forming the displacement loop or D-loop. Engaging the second end of the DSB mediated by Rad52 forms a double Holliday junction. The double Holliday junction is resolved by the combined action of the helicase Sgs1, the topoisomerase Top3, or Mus81 and Mms4, resulting in crossover and non-crossover products.

**Figure 7 ijms-23-11665-f007:**
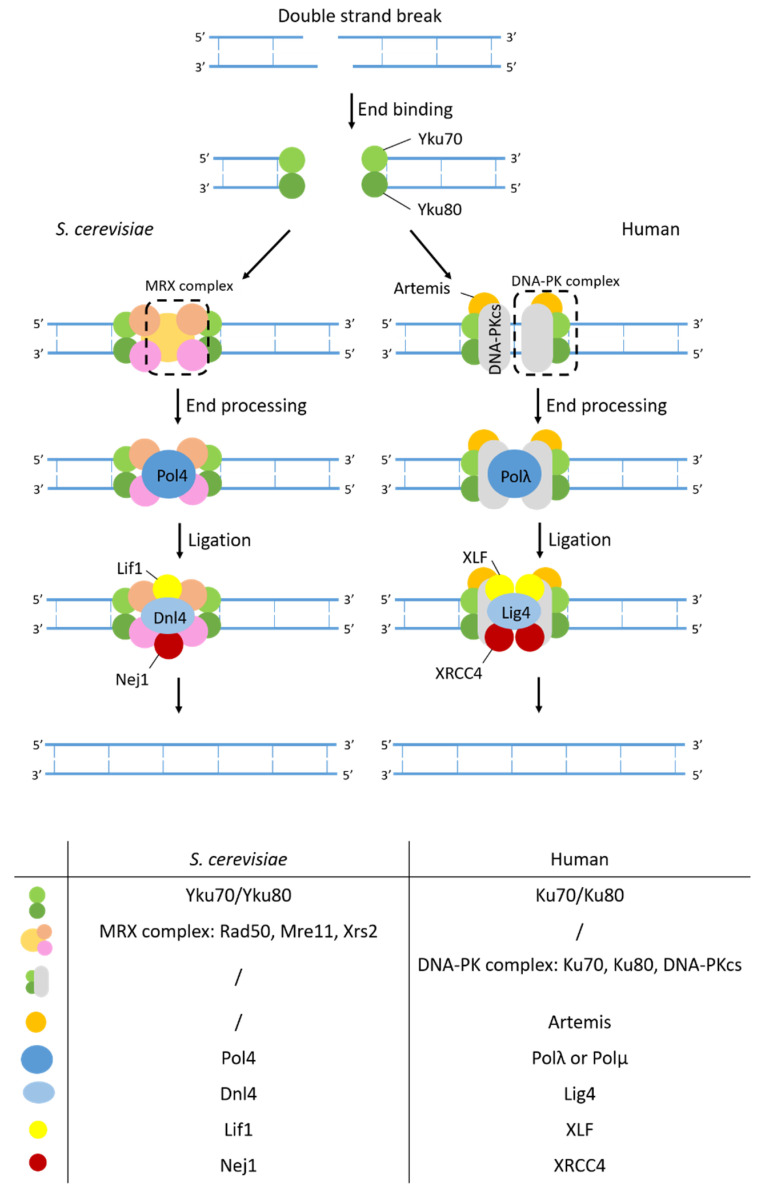
Mechanism of NHEJ. In yeast NHEJ, Yku70 and Yku80 bind the DSB ends together with the MRX complex. The Ku complex recruits other NHEJ proteins such as ligase Dnl4-Lif1 and the stabilizing factor Nej1 forming the repair super complex. Pol4 works together with the repair super complex making complementary ends which are ligated by Dnl4. In humans the process of NHEJ is largely similar, although the MRX complex does not work in NHEJ. Instead, this role is taken up by DNA-PK. Human homologs of the different yeast proteins are listed in the table.

## Data Availability

Not applicable.

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
