# Peer review of "Saccharomyces cerevisiae* as a Model System for Eukaryotic Cell Biology, from Cell Cycle Control to DNA Damage Response"

_ijms, 2022, doi:10.3390/ijms231911665_

Round 1
Reviewer 1 Report
I have now carefully read the manuscript presented by Vanderwaeren et al. The paper is well written and the diagrams are well presented. However I have a few concerns listed below that I suggest are dealt with:
1. More than half of the text reviews DNA damage pathways while the main topic of this paper is the crucial role that S.cerevisiae played in the past as a model organism to study genome stability mechanisms. In my opinion this thread should be expanded otherwise this paper will be another overview of DNA damage repair mechanisms.
2. I feel that the role of S.cerevisiae as a leading organism to study genome maintenance mechanisms in the Crispr/CAS9 era requires more discussion. What is the future of S.cerevisiae in this field?
3. “Signaling of DNA damage” chapter needs to be rewritten. The title should be changed as the chapter refers to DNA double strand breaks only and not to ssDNA gaps or stalled/damaged replication forks that induce different checkpoint pathways. Next, Tel1 is not only activated but also recruited by MRX. Finally, Tel1 is able to phosphorylate not only Chk1 but also Rad9 and Rad53.
Reviewer 2 Report
In the present manuscript, authors have tried to explain the characteristics of yeast as a model for DNA damage response and cell cycle control. Yeast and humans have highly conserved DNA damage response, and a number of yeast proteins have human homologs. It is an interesting topic, which shows how yeast is an ideal model for DNA damage study. The review is well organized and written in a clear manner.
In this review, all the yeast repair mechanisms are explained in detail, and their human homologs are also mentioned but there are number of reviews on repair pathways of yeast and their similarity with humans. To make this review more resourceful, it would be interesting if the authors explain about the studies or tools to use yeast system that will help in studying the repair mechanism. Such as recently, in 2020, a study presented a new tool to study DSB repair at a local and genome-wide scale in Saccharomyces cerevisiae. They generated a heterologous induction system to induce DSBs at one or multiple defined loci in the yeast genome using bacterial restriction enzymes. This system is media-independent, tightly regulated, and efficient DSB formation. Here is the link for paper: doi: 10.1093/nar/gkaa833
Minor changes:
In line 364, it should be HR as you have already mentioned the full form on page 1.
In figure 7, Protein names are only mentioned in the table. Label the proteins in same way as you did in other figures. Also, use a different bright color for RPA protein.
